# PSMA-PET Guided Treatment in Prostate Cancer Patients with Oligorecurrent Progression after Previous Salvage Treatment

**DOI:** 10.3390/cancers15072027

**Published:** 2023-03-29

**Authors:** Lorenzo Bianchi, Francesco Ceci, Eleonora Balestrazzi, Francesco Costa, Matteo Droghetti, Pietro Piazza, Alessandro Pissavini, Massimiliano Presutti, Andrea Farolfi, Riccardo Mei, Paolo Castellucci, Giorgio Gandaglia, Alessandro Larcher, Daniele Robesti, Alexandre Mottrie, Alberto Briganti, Alessio Giuseppe Morganti, Stefano Fanti, Francesco Montorsi, Riccardo Schiavina, Eugenio Brunocilla

**Affiliations:** 1Division of Urology, IRCCS Azienda Ospedaliero Universitaria di Bologna, 40138 Bologna, Italy; 2Department of Medical and Surgical Sciences, University of Bologna, 40127 Bologna, Italy; 3Division of Nuclear Medicine, IEO European Institute of Oncology IRCCS, 20141 Milan, Italy; 4Department of Oncology and Hemato-Oncology, University of Milan, 20122 Milan, Italy; 5Nuclear Medicine Division, IRCCS Azienda Ospedaliero Universitaria di Bologna, 40138 Bologna, Italy; 6Unit of Urology, Division of Experimental Oncology, Urological Research Institute, IRCCS San Raffaele Scientific Institute, Vita-Salute San Raffaele University, 20132 Milan, Italy; 7Department of Urology, Onze-Lieve-Vrouwziekenhuis, 9300 Aalst, Belgium; 8ORSI Academy, 9090 Melle, Belgium; 9Radiation Oncology Division, IRCCS Azienda Ospedaliero Universitaria di Bologna, 40138 Bologna, Italy

**Keywords:** PSMA-PET, hormone sensitive prostate cancer, oligorecurrent prostate cancer, metastasis-directed therapy, survival

## Abstract

**Simple Summary:**

Prostate Specific Membrane Antigen-Positron Emission Tomography (PSMA-PET) is currently recommended to stage Prostate Cancer (PCa) patients with recurrent disease and to select patients for metastasis-directed therapy (MDT). We aimed to evaluate the oncologic outcomes of second-line PSMA-guided MDT in oligo-recurrent PCa patients. Patients with oligorecurrent PCa (≤3 lesion in N1/M1a-b) who underwent MDT had similar progression compared to the conventional approach. However, individuals referred to MDT had a significantly lower risk of metastases and a lower risk of experiencing Castration Resistant Pca (CRPC) disease compared to those who were treated via the conventional approach. In patients undergoing MDT, no significant differences were found for risk of progression and metastasis according to N1 vs. M1a-b disease, while patients with M1a-b disease had higher risk of developing CRPC disease compared to those with N1 at PSMA-PET.

**Abstract:**

Background: Prostate Specific Membrane Antigen-Positron Emission Tomography (PSMA-PET) is used to select recurrent prostate cancer (PCa) patients for metastases-directed therapy (MDT). We aimed to evaluate the oncologic outcomes of second-line PSMA-guided MDT in oligo-recurrent PCa patients. Methods: we performed a retrospective analysis of 113 recurrent PCa after previous radical prostatectomy and salvage therapies with oligorecurrent disease at PSMA-PET (≤3 lesions in N1/M1a-b) in three high-volume European centres. Patients underwent second-line salvage treatments: MDT targeted to PSMA (including surgery and/or radiotherapy), and the conventional approach (observation or Androgen Deprivation Therapy [ADT]). Patients were stratified according to treatments (MDT vs. conventional approach). Patients who underwent MDT were stratified according to stage in PSMA-PET (N1 vs. M1a-b). The primary outcome of the study was Progression-free survival (PFS). Secondary outcomes were Metastases-free survival (MFS) and Castration Resistant PCa free survival (CRPC-FS). Kaplan-Meier analyses assessed PFS, MFS and CRPC-FS. Multivariable Cox regression models identified predictors of progression and metastatic disease. Results: Overall, 91 (80%) and 22 (20%) patients were treated with MDT and the conventional approach, respectively. The median follow-up after PSMA-PET was 31 months. Patients who underwent MDT had a similar PFS compared to the conventional approach (*p* = 0.3). Individuals referred to MDT had significantly higher MFS and CRPC-FS compared to those who were treated with the conventional approach (73.5% and 94.7% vs. 30.5% and 79.5%; all *p* ≤ 0.001). In patients undergoing MDT, no significant differences were found for PFS and MFS according to N1 vs. M1a-b disease, while CRPC-FS estimates were significantly higher in patients with N1 vs. M1a-b (100% vs. 86.1%; *p* = 0.02). At multivariable analyses, age (HR = 0.96) and ADT during second line salvage treatment (HR = 0.5) were independent predictors of PFS; MDT (HR 0.27) was the only independent predictor of MFS (all *p* ≤ 0.04) Conclusion: Patients who underwent second-line PSMA-guided MDT experienced higher MFS and CRPC-FS compared to men who received conventional management.

## 1. Introduction

The clinical management of recurrent prostate cancer (PCa) changed significantly after the introduction of a new generation of imaging. Modern diagnostic procedures identify patients with oligometastatic disease [1] earlier and with more accuracy compared to conventional imaging. Prostate Specific Membrane Antigen/Positron Emission Tomography (PSMA-PET) represents the gold standard [2] in cases of biochemical recurrence (BCR) due to its high accuracy in correctly detecting and localizing lesions, [3] namely in the early stage of recurrence, with low prostate specific antigen (PSA) levels [4]. Therefore, there is increasing interest in metastasis-directed therapy (MDT) that is guided by new generation imaging. The objective of MDT is to treat all visible PCa metastases to prevent or delay further metastatic spread, potentially improving patient outcomes [5] compared to conventional approaches (usually observation or systemic treatments such as androgen deprivation therapy [ADT]). The ORIOLE trial [6] (a phase 2 randomized study in which oligometastatic PCa patients were randomized to receive Stereotactic Body RadioTherapy [SBRT] or observation with disease progression as primary outcome) and STOMP [1] trial (a phase 2 randomized study in which oligometastatic PCa patients were randomized to receive MDT of all detected lesions [surgery or SBRT] or observation with ADT-free survival as the primary outcome) proved the safety and feasibility of MDT in these settings by delaying the administration of ADT [1] and disease progression [6]. Nevertheless, the efficacy of MDT as a curative treatment in oligo-recurrent PCa is still under debate. Thus, patients need to be accurately selected, and MDT efficacy is associated with the correct identification of all metastatic sites [7]. PSMA-PET may be a prognostic tool for recurrent PCa [8,9], and has the potential to be the optimal procedure for image-guided MDT, as was recently proposed in the PEACE V-STORM trial [10] (a phase 2 randomized study in which nodal pelvic oligorecurrent PCa patients based on PET imaging were randomized to receive MDT+ 6 months of ADT or whole pelvic radiotherapy + MDT + 6 months of ADT with metastasis-free survival as the primary outcome). Moreover, oligorecurrent PCa represents a heterogenous group of men with different patterns of disease recurrence, including patients with biochemical persistence (BCP), BCR at first presentation, and further PSA recurrence after salvage therapies, who are in the late phase of the PCa natural history with a lower chance of being cured.

Thus, we aim to explore the oncologic outcomes of second-line PSMA-guided MDT in PCa patients with PSA progression after previous salvage treatments and oligo-recurrence detected with PSMA-PET.

## 2. Materials and Methods

### 2.1. Study Design and Population Characteristics

This study enrolled patients through a multicenter collaboration among three tertiary high-volume European centers (IRCCS Sant’ Orsola-Malpighi in Bologna, IRCCS San Raffaele in Milan, and the OLV Hospital in Aalst). In all centers, patients were included in accordance with Institutional Review Board (IRB) and ethical committee approvals (Prot. PSMA-PROSTATA; Eudract: 2015-004589-27 OsSC) and signed informed consent forms (ICF), as per local requirements. The clinical records of PCa patients who performed RP between January 1998 and January 2021 and PSMA-PET from January 2016 and February 2021 were retrospectively analyzed. Inclusion criteria were: (1) proven PCa; (2) hormone-sensitive PCa (HSPC) and ADT-free at the time of PSMA-PET (for at least 6 months); (3) PSA failure after previous salvage therapies (i.e., prostate bed radiotherapy [RT], whole pelvis RT, whole pelvis RT + ADT or ADT alone); (4) PSMA-PET scan performed during PSA relapse; (5) oligorecurrent disease at PSMA-PET (defined as ≤3 lesions [11] in N1/M1a-b); (6) patients eligible for MDT. Patients showing local recurrence only and/or visceral metastases (M1c) were excluded from this analysis. Patients with biochemical persistence (n = 10), BCR after surgery who never received salvage therapies (53), patients who received previous chemotherapy or androgen receptor targeted agents (ARTA; n = 7), and Castration resistant Prostate Cancer (CRPC) patients (n = 20) at the time of the PSMA-PET were also excluded. A total of 113 (n = 113) patients who met the inclusion/exclusion criteria with a minimum follow up of 12 months after PSMA-PET were considered eligible for primary end-point analysis (Figure 1).

### 2.2. PSMA-PET Procedure and Interpretation Criteria

68Ga-PSMA-11 was synthesized in all involved centers according to good manufacturing practices (GMP) and in accordance with international procedural guidelines [12,13]. A mean dose of 1.8–2.2 MBq/Kg body weight of 68Ga-PSMA-11 was administered intravenously. 68Ga-PSMA-11 PET/Computed Tomography (CT) was performed with a standard technique, and was in accordance with international procedural guidelines [14]. All studies were performed using dedicated PET/CT state-of-the-art scanners. All PSMA-PET images were locally reviewed independently and with the of two experienced nuclear medicine physicians according to international reporting guidelines [15,16]. In cases of disagreement, consensus was reached by involving a third reader (the 2:1 rule).

### 2.3. Patient Management and Treatments

PSMA-PET findings have been interpreted according to international procedural guidelines [17]. Patients were staged according to molecular imaging TNM (miTNM) [18], taking into account the PSMA-PET findings. Therapies after PSMA-PET have been administered according to international urologic guidelines [19], and treatment management decisions were made by a multidisciplinary tumor board, considering previous treatments and patient preference [20]. In brief, patients underwent second-line salvage treatments that consisted of either the MDT approach targeted to PSMA positive lesions according to relapse patterns (including salvage lymphadenectomy (sLND, n = 38), SBRT targeted to nodal (n = 20) or skeletal lesions (n = 31), or a combination of sLND and SBRT (n = 2)), or the conventional approach (including observation (n = 2) or ADT, (n = 20)). Patients were stratified according to type of salvage treatments (MDT vs. conventional approach). Short term ADT was allowed as an adjuvant treatment after MDT according to international procedural guidelines [18], multidisciplinary tumor board decisions, and patient preference.

### 2.4. Outcomes Measurement

The primary outcome of the study was Progression-free survival (PFS), defined as the time in months between the date of PSMA-PET and the date of progression or last follow-up. Progression was defined as one of the following: (a) PSA progression; (b) radiological progression, defined as the appearance of new PCa localization(s) at any imaging procedure (PSMA-PET and/or Choline-PET and/or whole-body MRI) performed during follow-up; and (c) death due to any cause.

Secondary outcomes were Metastases-free survival (MFS), defined as the appearance of new PCa metastases at any imaging procedure performed during follow-up, and CRPC-free survival (CRPC-FS), defined as the occurrence of CRPC [19] (both metastatic and non-metastatic) during follow-up.

### 2.5. Statistical Analyses

Statistical analyses firstly consisted of descriptive statistics in the overall population, stratifying according to the therapy strategy performed after PSMA-PET (MDT vs. conventional approach). The Chi-squared and the Mann–Whitney tests were used to compare proportions and medians between the two groups, respectively.

Second, Kaplan–Meier analyses were used to assess PFS, MFS and CRPC-FS estimates at 3 years follow-up in the overall population, stratifying according to treatment after PSMA-PET (MDT vs. conventional approach) and compared by the log-rank test. Moreover, in patients who underwent the MDT approach, PFS, MFS and CRPC-FS estimates at 3 years follow-up were stratified according to stage in PSMA-PET (mi N1 vs. mi M1a-b).

Third, a multivariable Cox regression was performed to identify independent predictors of progression and metastases. Covariates were age, pathologic ISUP group, ADT during second-line salvage treatment, miTNM stage [21] (namely, N1 vs. M1a-M1b), number of positive lesions at PSMA-PET, and type of salvage approach (MDT vs. the conventional approach).

All statistical tests were performed with R 4.0.3 (R Foundation for Statistical Computing, Vienna, Austria) with a 2-sided significance level set at *p* < 0.05.

## 3. Results

Baseline characteristics of the overall population are reported in Table 1. Overall, 42 (37.2%), 52 (46%), 3 (2.7%) and 16 (14.2%) patients experienced PSA recurrence after previous salvage prostate bed RT, whole pelvis RT, whole pelvis RT + ADT, and ADT only, respectively. The median PSA at PSMA-PET was 0.62 ng/mL.

Overall, 91 out of 113 (80%) patients were treated with MDT and 22 out 113 (20%) men underwent the conventional approach. The median (IQR) number of positive lesions at PSMA-PET was 1 (1–2) and 2 (1–2) in patients treated with MDT and the conventional approach, respectively (*p* = 0.04). Overall, 58.4% and 41.6% of patients had miN1 and miM1a-b disease at PSMA-PET, with no significant differences between the two groups (*p* = 0.06; Table 2).

Figure 2 shows the distribution of positive PSMA-PET recurrence according to miTNM.

The median (IQR) follow-up from RP and PSMA-PET was 52 months (30–94), and after PSMA-PET it was 31 (19–42) months. According to the Kaplan-Meier curve, the 3-years PFS estimates were 41.8% and 13.8% in patients who underwent MDT and the conventional approach, respectively (*p* = 0.3, Figure 3a). At 3-years follow up, the MFS estimates were significantly higher in patients undergoing MDT compared to men treated with the conventional approach (73.5% vs. 30.5%, *p* < 0.001; Figure 3b). In the overall population, the CRPC-FS estimates at 3 years were significantly higher in patients who underwent MDT compared to men treated with the conventional approach (94.7% vs. 79.5%, *p* < 0.001; Figure 3c).

In patients who underwent MDT, no significant differences were found for PFS and MFS estimates stratifying the population according to N1 vs. M1a-b disease (38.9% vs. 46.1% and 77.4% vs. 67% at 3 years; *p* ≥ 0.7; Figure 4a,b). However, the CRPC-FS estimates at 3 years were significantly higher in patients with N1 vs. M1a-b localization (100% vs. 86.1%; *p* = 0.02; Figure 4c).

Finally, at the multivariable Cox regression analysis, age (HR = 0.96; 95% CI: 0.92–0.99) and ADT during second line salvage treatment (HR = 0.50; 95% CI: 0.27–0.93) were independent predictors of PFS (all *p* ≤ 0.04), while the use of MDT (HR 0.27; 95% CI: 0.10–0.69) proved to be the only independent predictor of MFS (*p* = 0.006; Table 3).

## 4. Discussion

PSMA-PET is considered the imaging method of choice to identify PCa lesions in patients with recurrent PCa after primary treatments [19,22,23,24]. Thus, considering patients who show PSA recurrence after salvage therapies, PSMA-PET may identify oligo-recurrent men eligible for MDT. Two randomized phase II trials [1,6] supported the use of MDT in recurrent PCa patients, showing more favorable outcomes compared to the conventional approach by delaying the administration of ADT and disease progression. However, in the STOMP trial [1,25], the selection of oligometastatic patients was based on Choline-PET, while men were enrolled by conventional imagine in the ORIOLE trial [6]. Despite this, in a post-hoc analysis of the ORIOLE trial, total consolidation of PSMA-positive disease decreased the risk of new lesions at 6 months (16% vs. 63%; *p* = 0.006). Moreover, most patients who underwent MDT in the ORIOLE and STOMP trials would be ineligible whenever PSMA-PET was used for patient selection, and the effect of MDT would be different. Indeed, the treatment of small metastases detected by PSMA-PET could be more effective in terms of oncologic survival compared to the treatment of larger lesions detected by less sensitive imaging techniques.

Thus, the introduction of PSMA-PET in a biochemical recurrent setting generated a stage migration towards metastatic HSPC, leading to different therapeutic scenarios [26,27]. It is worthy of note that the identification of oligometastatic disease by PSMA-PET may select PCa patients for MDT in an earlier stage with a potentially greater chance of being cured. Holscher T et al. [28] showed that up to five metastases identified by PSMA-PET in metachronous progressing PCa patients can safely be targeted by local ablative RT as part of MDT. However, long-term studies of oncologic outcomes of treatments changed by guided PSMA PET do not exist.

In this multicentric study, we included HSPC patients with oligorecurrent disease (1–3 lesions in PSMA-PET) suitable for MDT. The oncological outcomes of PSMA-guided MDT have been evaluated. There is no concordance regarding the definitions of oligometastatic and oligo-recurrent disease [5]. In our study we analyzed oligo-recurrent patients with ≤3 lesion (N1 and/or M1a-b), and thus patients with a lower disease burden. Moreover, since oligo-recurrent PCa represent a heterogenous group of men (BCP, first-time BCR, PSA failure after salvage therapy), we aimed to explore the potential benefit of PSMA-guided MDT as a second-line salvage treatment in patients who already received previous first-line salvage therapies for PSA relapse. The management of patients in the post-salvage setting is challenging indeed, since patients had already received different cancer treatments (both primary and subsequent salvage therapies), limiting the therapeutic chances, and with the risk of progression to CRPC being higher.

In our cohort, the median time from RP to PSA recurrence after salvage therapy was 52 months, and patients had a median PSA level of 0.6 ng/mL at the time of PSMA-PET. We observed that patients treated with PSMA-guided MDT experienced similar PFS rates compared to men who underwent the conventional approach (Figure 3). As reported by Bravi et al. in a large multicentric experience [29] on sLND for nodal recurrent PCa, MDT alone was associated with durable long-term outcomes in a minority of men who significantly benefited from additional and combined treatments. Moreover, recent prospective data on SBRT targeted to oligo-recurrent PCa patients after radical prostatectomy and postoperative radiotherapy showed that only the 20% of patients receiving MDT alone had no biochemical evidence of disease, with an overall response rate of 60%. Accordingly, in our population, 53.8% of patients had a PSA relapse after PSMA-guided MDT. In those patients who underwent ablative SBRT for nodal or bone metastases (n = 63), the 25% experienced immediate PSA progression [28]. This could be due to the presence of residual micro-metastases that remain undetectable even with PSMA-PET. Indeed, PSMA-PET allows for the detection of the “tip of the iceberg” or the most relevant lesions that could be treated by MDT, leading to two hypotheses: one, patients achieve immediate response because of micro-metastases outside the MDT target not detected by PSMA-PET that remain silent before further progression; and two, patients have no complete response due to micro-metastases not visible at PSMA-PET (i.e. not treated with MDT) that are still “active” [30].

However, at a median follow up of 31 months after PSMA-PET, patients treated with PSMA directed MDT had significantly higher MFS (73.5%) and CRPC-FS (94.7%) rates compared to men who underwent the conventional approach (30.5% and 79.5%, respectively). These results are promising, as we observed a significant lower risk of CRPC progression in patients treated with the PSMA-guided MDT approach, even if patients mainly received ADT in the control group. Thus, in a selected population of high-risk men with oligorecurrent PCa, the PSMA-PET directed approach may represent a promising second-line salvage approach to delay the further progression to polimetastatic and CRPC status. Further studies investigating the combination of systemic therapies with novel ARTA and MDT are required in order to evaluate the oncologic benefit in the setting of oligorecurrent HSPC. Accordingly, the adoption of second-line PSMA-guided MDT was the only independent predictor of MFS after adjusting for the use of ADT at the time of salvage therapy, miTNM stage and number of positive lesions at PSMA-PET. This hypothesis is supported by previous evidence in the ORIOLE trial [6], in which the consolidation of all macroscopic metastases may remove or significantly affect signals that promote the development of remaining micro-metastases, suggesting that MDT could be a potential curative therapy in selected oligometastatic men.

Stratifying our population according to the disease stage identified in PSMA-PET (N1 vs. M1 disease), no significant differences were found for PFS and MFS. Thus, patients with a limited number of nodal and/or skeletal lesions might be referred to PSMA-guided MDT. However, the CRPC-FS rates were significantly higher in patients who underwent MDT for N1 localization at PSMA-PET, suggesting that more favorable outcomes might be achieved in men with limited pelvic nodal disease.

### Limitations

This study is not exempt from limitations. First, the retrospective design of the study might have generated issues regarding patient selection. However, these data have been derived in each center by prospective studies in consecutive patients. Second, even if a central review was not performed, all PSMA-PET images were evaluated with a local review by PSMA-PET experienced nuclear medicine physicians according to international reporting procedural guidelines. Third, the short follow-up time after PSMA limited further consideration of long-terms outcomes. Fourth, the histologic validation of positive findings was not feasible in all cases due to ethical and practical reasons, and thus the presence of false positive findings cannot be excluded. Fifth, the potential benefit of MDT targeted to PSMA lesions compared to conventional treatment may be limited by the low number of patients who underwent the conventional approach and the inherent differences between the two groups. Indeed, several biases due to different sRT protocols, different sites, and the extent of disease on PSMA-PET, different types of MDT ranging from only surgery or radiotherapy, to combination treatment to systemic ADT, or even observation need be highlighted, and final conclusions on this topic should be taken with caution. A prospective randomized trial would have been preferable to this one. However, this study provides evidence from a real-world scenario. Finally, no direct comparison of different MDT approaches (surgery vs. SBRT vs. combination) has been evaluated due to the inclusion of different sites of recurrent lesions (both nodal and skeletal) that are suitable for MDT.

## 5. Conclusions

In PCa patients with oligo-recurrent disease after previous salvage treatments, PSMA-PET could be used to personalize second-line salvage therapies by adopting an MDT approach. Patients who underwent PSMA-guided MDT experienced similar PFS and higher MFS and CRPC-FS compared to men who received conventional management. Thus, in a selected population of high-risk men with oligo-recurrent N1/M1a-b disease and limited therapeutic chances due to previous salvage treatments, the PSMA-PET directed approach with the consolidation of metastatic lesions may represent a promising second-line salvage approach to delay the further progression of disease to CRPC status.

## Figures and Tables

**Figure 1 cancers-15-02027-f001:**
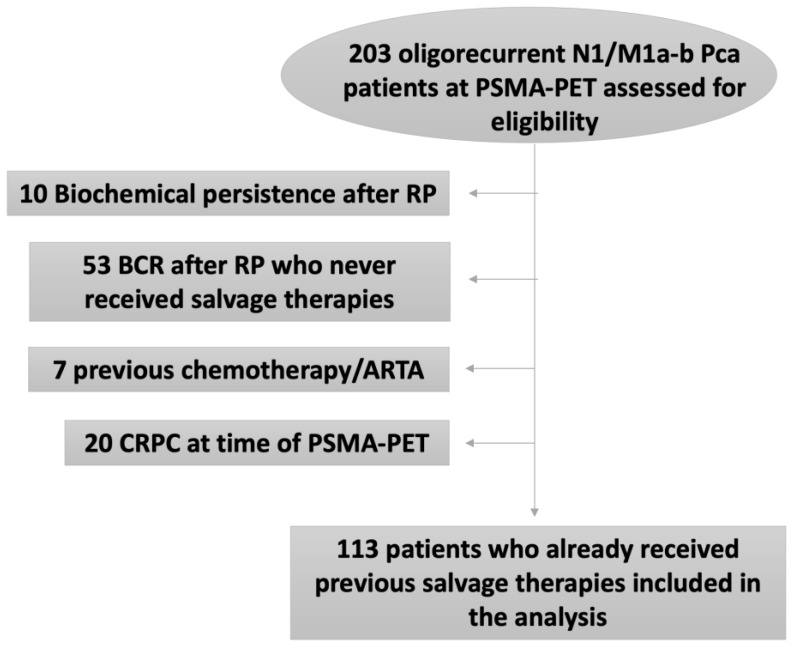
Flow chart of participants in the study.

**Figure 2 cancers-15-02027-f002:**
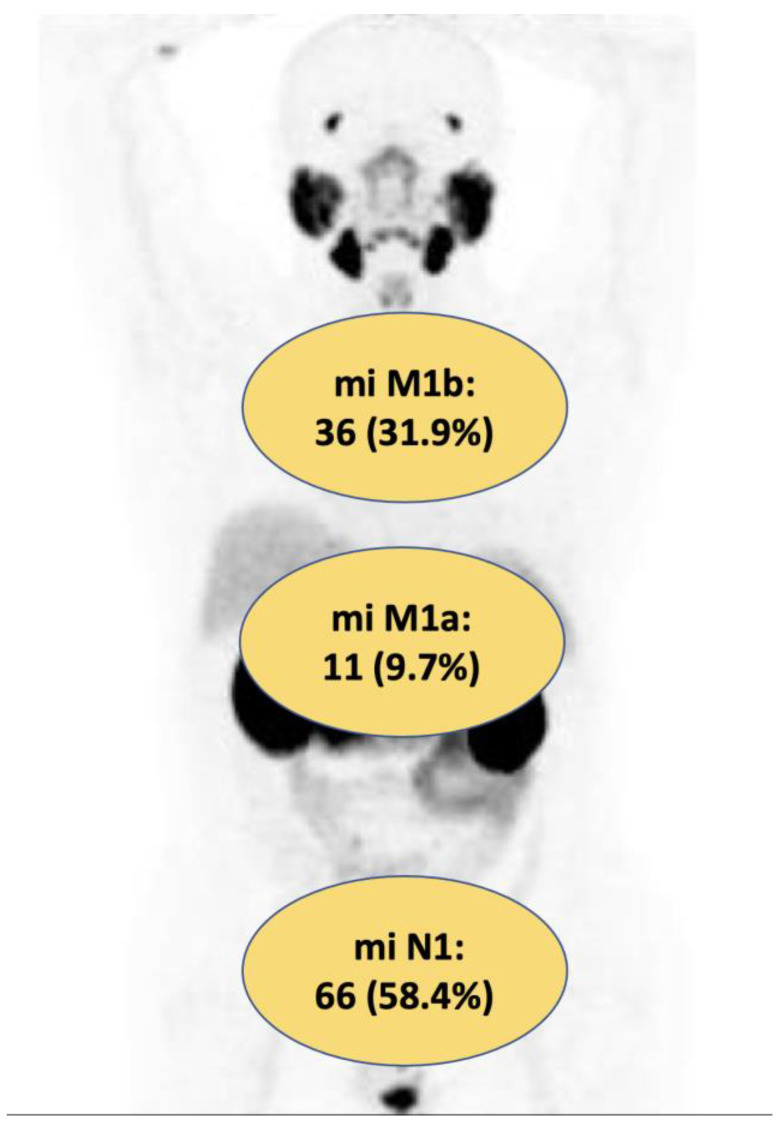
Localization of oligorecurrent disease detected by PSMA-PET according to miTNM.

**Figure 3 cancers-15-02027-f003:**
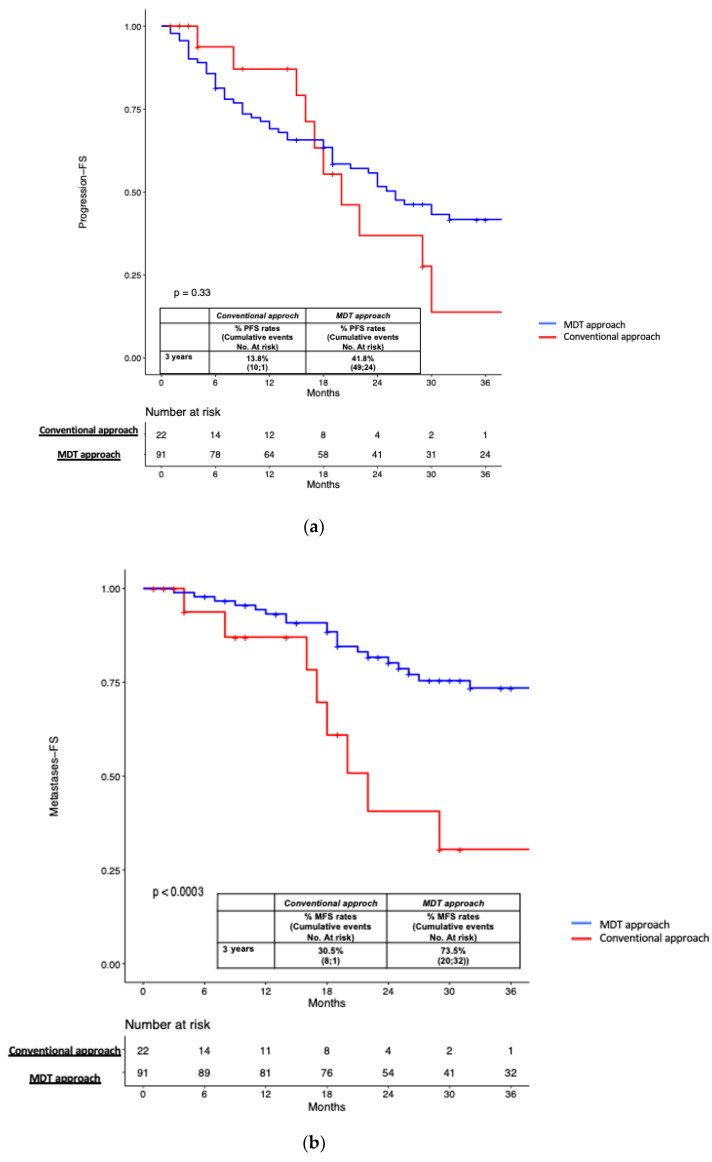
(**a**) Kaplan-Meier curve depicting Progression Free Survival (PFS) rates in the overall population (n = 113) according to type of treatment performed (MDT approach vs. conventional approach); (**b**) Kaplan-Meier curve depicting Metastases Free Survival (MPFS) rates in the overall population (n = 113) according to type of treatment performed (MDT approach vs. conventional approach); (**c**) Kaplan-Meier curve depicting Castration Resistant Prostate Cancer Free Survival (CRPC-FS) rates in the overall population (n = 113) according to type of treatment performed (MDT approach vs. conventional approach).

**Figure 4 cancers-15-02027-f004:**
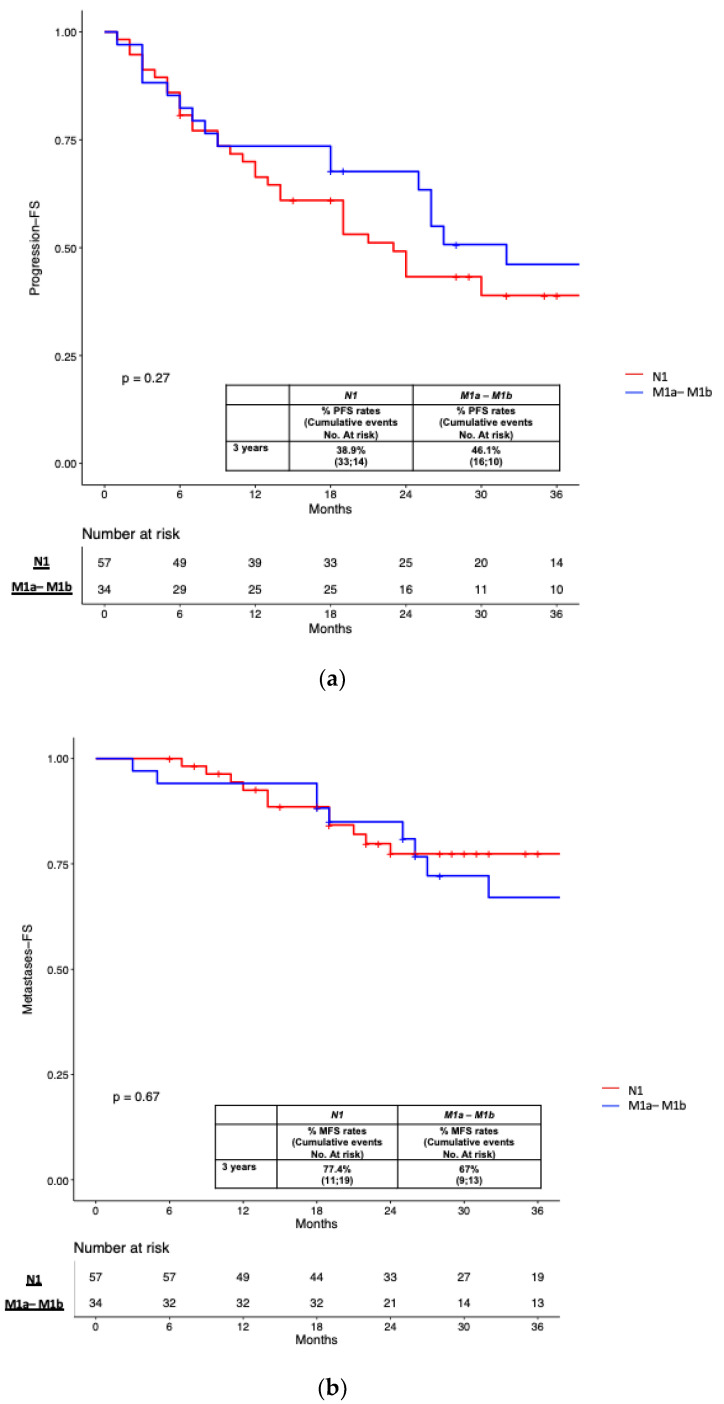
(**a**) Kaplan-Meier curve depicting Progression Free Survival (PFS) rates in patients treated with the MDT approach (n = 91) according to miTNM stage at PSMA-PET (miN1 vs. miM1a-M1b); (**b**) Kaplan-Meier curve depicting Metastases Free Survival (PFS) rates in patients treated with the MDT approach (n = 91) according to miTNM stage at PSMA-PET (miN1 vs. miM1a-M1b); (**c**) Kaplan-Meier curve depicting Castration Resistant Prostate Cancer Free Survival (CRPC-FS) rates in patients treated with the MDT approach (n = 91) according to miTNM stage at PSMA-PET (miN1 vs. miM1a-M1b).

**Table 1 cancers-15-02027-t001:** Overall descriptive characteristics in the overall population (n = 113).

	Overall
Patients, n (%)	113 (100)
Age	
Median (IQR)	61 (56–66)
pT stage, n (%)	
pT2	38 (33.6)
pT3a	37 (32.7)
pT3b–pT4	38 (33.6)
pN stage, n (%)	
pNx	30 (26.5)
pN0	58 (51.3)
pN1	25 (22.1)
Pathologic ISUP grade, n (%)	
1–3	61 (54)
4–5	52 (46)
Salvage therapies, n (%)	
Prostate Bed RT	42 (37.2)
Whole Pelvis RT	52 (46)
Whole Pelvis RT + ADT	3 (2.7)
ADT	16 (14.2)
PSA level at PSMA-PET, ng/mL	
Median (IQR)	0.62 (0.29–1.27)
Time from RP to PSMA-PET, months	
Median (IQR)	52 (30–94)
Follow-up after PSMA-PET, months	
Median (IQR)	31 (19–42)

ISUP: International Society of Urological Pathology; IQR: interquartile range; PSA: prostate specific antigen; RT: radiotherapy; ADT: Androgen deprivation therapy; RP: radical prostatectomy; PSMA: Prostate Specific Membrane Antigen; PET: Positron Emission Tomography.

**Table 2 cancers-15-02027-t002:** PSMA-PET results and oncologic outcomes after PSMA-guided salvage treatments stratified according to type of treatment (namely, MDT vs. conventional approach).

	Overall	MDT Approach	Conventional Approach	*p* Value
Patients, n (%)	113 (100)	91 (80)	22 (20)	-
Number of positive lesions at PSMA-PET, n				0.04
Median (IQR)	1 (1–2)	1 (1–2)	2 (1–2)	
Site of positive PSMA-PET, n (%)				0.1
LND	77 (68.1)	65 (71.4)	12 (54.5)	
Bones	24 (21.2)	19 (20.9)	5 (22.7)	
LND and Bones	12 (10.6)	7 (6.2)	5 (22.7)	
mi stage at PSMA-PET, n (%)				0.06
N1	66 (58.4)	57 (62.6)	9 (40.9)	
M1a-b	47 (41.6)	34 (37.4)	13 (59.1)	
Progression after PSMA-PET guided treatment, n (%)				0.3
No	45 (39.8)	34 (37.4)	11 (50)	
Yes	68 (60.2)	57 (62.6)	11 (50)	
Time to progression, months				0.012
Median (IQR)	16 (6–26)	13 (6–25)	18 (15–29)	
PSA recurrence after PSMA-PET guided treatment, n (%)				0.5
No	54 (47.8)	42 (46.2)	12 (54.5)	
Yes	59 (52.2)	49 (53.8)	10 (45.5)	
PSA at recurrence after PSMA-PET guided treatment, ng/mL				0.8
Median (IQR)	3.18 (0.7–14)	3.4 (0.69–1.5)	2.54 (0.95–16.25)	
Metastases after PSMA-PET guided treatment, n (%)				0.1
No	81 (71.7)	68 (74.7)	13 (59.1)	
Yes	32 (28.3)	23 (25.3)	9 (40.9)	
Time to Metastases, months				0.8
Median (IQR)	19 (13–26)	19 (12–26)	18 (12–26)	
CRPC after PSMA-PET guided treatment, n (%)				0.03
No	98 (86.7)	82 (90.1)	16 (72.7)	
Yes	15 (13.3)	9 (9.9)	6 (27.3)	
Time to CRPC, months				≤0.001
Median (IQR)	37 (18–43)	37 (19–42)	29 (4–43)	
Overall Mortality, n (%)				0.3
No	108 (95.6)	86 (94.5)	22 (100)	
Yes	5 (4.4)	5 (5.5)	0 (0)	
Cancer specific mortality, n (%)				0.5
No	111 (98.2)	89 (97.8)	22 (100)	
Yes	2 (1.8)	2 (2.2)	0 (0)	

MDT: metastasis directed treatment; PSMA: Prostate Specific Membrane Antigen; PET: Positron Emission Tomography; LND: lymph-nodal; sLND: salvage lymph node dissection; SBRT: stereotactic body radiotherapy; sRT: salvage radiotherapy; ADT: androgen deprivation therapy; IQR: interquartile range; CRPC: castration resistant prostate cancer.

**Table 3 cancers-15-02027-t003:** Multivariate Cox regression to predict Progression free survival and Metastasis free survival in the overall population (n = 113).

Variables	Progression		Metastasis	
	HR (95% C.I.)	*p* Value	HR (95% C.I.)	*p* Value
Age (years)	0.96 (0.92–0.99)	0.04	-	-
Pathologic ISUP group			-	-
ISUP 1–3	1–0 (Ref)	0.9		
ISUP 4–5	1.04 (0.57–1.87)			
miTNM stage				
N1	1–0 (Ref)	0.8	1–0 (Ref)	0.9
M1a-M1b	1.06 (0.55–2.06)		0.97 (0.43–2.20)	
Number of positive lesions at PSMA-PET	0.98 (0.67–1.439)	0.9	1.03 (0.60–1.80)	0.9
ADT during second line salvage treatment				
No	1–0 (Ref)		1.0 (Ref)	
Yes	0.50 (0.27–0.93)	0.03	1.95 (0.82–4.62)	0.1
Type of salvage treatment after PSMA-PET				
Conventional approach	1.0 (Ref)		1.0 (Ref)	
MDT	0.49 (0.20–1.25)	0.1	0.27 (0.10–0.69)	0.006

ISUP: International Society of Urological Pathology; HR: Hazzard ratio; CI: confidence interval; PSMA: Prostate Specific Membrane Antigen; PET: Positron Emission Tomography.

## Data Availability

Not applicable.

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
