# Peer review of "PSMA-PET Guided Treatment in Prostate Cancer Patients with Oligorecurrent Progression after Previous Salvage Treatment"

_cancers, 2023, doi:10.3390/cancers15072027_

Round 1

Reviewer 1 Report

In this retrospective study, Bianchi et al. report on the use of PSMA PET/CT in personalizing 2nd line therapy approach, specifically the use of PSMA-targeted MDT vs. conventional therapy in patients with oligometastatic disease.  

I found the paper to be well written, the methodology to be sound, and the results and conclusions to have real potential clinical impact. The topic of oligometastatic disease in PCa is considered a "hot topic" and the use of PSMA imaging in this setting should be further studied.

Aside from some minor English language editing, I have no comments for the authors and I recommend to accept the paper in it's current form.

Author Response

Reviewer #1:

Aside from some minor English language editing, I have no comments for the authors and I recommend to accept the paper in it's current form.

We would like to thank the Reviewer for his/her comment and for the good opinion about our researches on this topic. We corrected some English language editing as suggested.

Reviewer 2 Report

The authors present a retrospective analysis of 113 patients with oligo-recurrent Prostate Cancer (n=113) after salvage therapies. The outcomes of PSMA-guided metastasis treatment (MDT) are compared with the standard procedure according tot he guidelines (oberservation or antiandrogenic therapy). With comparable progression-free intervals, the MDT cohort has higher metastasis-free and castration-resistant prostate cancer free intervals. It is concluded that PSMA-PET guided MDT is a promising option for second-line salvage therapy in high-risk patients with oligometastatic PCA to delay further disease progression to CRPC status.

The manuscript is well organized and the results are presented in a coherent manner. The results are placed in the context of recent literature (23 references between 2015-2022). The text is highly condensed and sophisticated. The work is mainly aimed at urologists involved in therapy, as very profound knowledge about the therapy regimes of metastatic prostate cancer is assumed. The results are presented in a scientifically sound manner in tables and figures. The conclusions are consistent with the results presented.

The paper represents a valuable contribution for future personalized therapies of metastatic prostate carcinoma. A reasoned proposal on the use of PSMA-PET is presented that extends beyond the recommendations of current guidelines. Limitations, particularly the relatively small subject population, are adequately discussed by the authors. I commend the publication of this work in Cancers but have minor suggestions for improvement for the authors:

General comments:

- The text is designed for specialists and is difficult to understand for a broader readership. I would recommend going into more detail about the conventional approach as opposed to PSMA-guided MDT. It would be helpful to briefly summarize the results of the cited studies (ORIOLE, STOMP, and PEACE V-STORM).

- In the introduction, it is mentioned that PSMA-guided MDT is even discussed as a possible curative therapy for oligometastatic prostate cancer. This issue could also be commented on in the discussion.

Specific comments:

- Title: superscript text at affliations.

- The abbreviations PSMA-PET, CRPC, and ADT are used for the first time in lines 30, 35, and 46 and were not previously explained.

- The miN1 and miM1a-b designations (line 190, especially the prefix mi) should be explained.

- The labels of figures 3 and 4 are much too small and not readable in a printout.

- Incorrect abbreviation in line 226: „metastases free survival (PFS)“.

- I think a relative pronoun is missing in sentence 262-265.

Author Response

Reviewer #2:

I commend the publication of this work in Cancers but have minor suggestions for improvement for the authors:

General comments:

- The text is designed for specialists and is difficult to understand for a broader readership. I would recommend going into more detail about the conventional approach as opposed to PSMA-guided MDT. It would be helpful to briefly summarize the results of the cited studies (ORIOLE, STOMP, and PEACE V-STORM).

We would like to thank the Reviewer for his/her comment. Conventional approach as opposed to MDT consist of observation or systemic treatment such as ADT. We implemented the manuscript accordingly adding short description of cited studies as follows:

-Page 3, lines 76-85: “The rational of MDT is to treat all visible PCa metastases to prevent or delay further metastatic spread and potentially improving patients’ outcomes [5] compared to conventional approach (usually observation or systemic treatments such as androgen deprivation therapy [ADT]). The ORIOLE trial [6] (phase 2 randomized study in which oligometastatic PCa patients were randomized to receive Stereotactic Body RadioTherapy [SBRT] or observation with disease progression as primary outcome) and STOMP [1] trial (phase 2 randomized study in which oligometastatic PCa patients were randomized to receive MDT of all detected lesions [surgery or SBRT] or observation with ADT-free survival as primary outcome) proved safety and feasibility of MDT in these settings by delaying the administration of ADT [1] and progression [6].

-Page 3, lines 89-94: “PSMA-PET may be a prognostic tool for recurrent PCa [8,9] and has the potential to be the optimal procedure for image-guided MDT, as recently proposed in the PEACE V-STORM trial [10] (phase 2 randomized study in which nodal pelvic oligorecurrent PCa patients based on PET imaging were randomized to receive MDT+ 6 months of ADT or whole pelvic radiotherapy + MDT + 6 months of ADT with metastasis-free survival as primary outcome).

- In the introduction, it is mentioned that PSMA-guided MDT is even discussed as a possible curative therapy for oligometastatic prostate cancer. This issue could also be commented on in the discussion.
We would like to thank the Reviewer for his/her comment. We implement discussion as follows:

-Page 14, lines 335-339: “This hypothesis is supported by previous evidence in ORIOLE trial6, in which the consolidation of all macroscopic metastases may remove or significantly affect signals that promote the development of remaining micro-metastases, suggesting that MDT could be a potential curative therapy in selected oligometastatic men”

Specific comments:

- Title: superscript text at affliations.

We would like to thank the Reviewer for his/her comment. We modified accordingly.

- The abbreviations PSMA-PET, CRPC, and ADT are used for the first time in lines 30, 35, and 46 and were not previously explained.

We would like to thank the Reviewer for his/her comment. We added explanation of abbreviation as suggested. Please see the modified texts as follows:

-Page 1, lines 30-32: “Prostate Specific Membrane Antigen-Positron Emission Tomography (PSMA-PET) is currently recommended to stage Prostate Cancer (PCa) patients with recurrent disease and to select patients for metastasis-directed therapy (MDT).”

-Page 1, lines 35-36: “However, individuals referred to MDT had significantly lower risk of metastases and to experience Castration Resistant Pca (CRPC) disease”

-Page 1, lines 46-47: “MDT targeted to PSMA (including surgery and/or radiotherapy) and conventional approach (observation or Androgen Deprivation Therapy [ADT])”.

- The miN1 and miM1a-b designations (line 190, especially the prefix mi) should be explained.

We would like to thank the Reviewer for his/her comment. miTNM refers to TNM based on molecular imaging findings (indeed PSMA-PET). We implemented the text accordingly. Please see the modified manuscript:

-Page 142, lines 129-130: “Patients were staged according to molecular imaging TNM (miTNM) [20], taking into account PSMA-PET findings.”

- The labels of figures 3 and 4 are much too small and not readable in a printout.

We would like to thank the Reviewer for his/her comment. We modified the labels of the figure 3 and 4 and modified the graphic dimensions.

- Incorrect abbreviation in line 226: „metastases free survival (PFS)“.

We would like to thank the Reviewer for his/her comment. We modified correctly PFS in MFS

- I think a relative pronoun is missing in sentence 262-265.

We would like to thank the Reviewer for his/her comment. We rephrased the sentence as follows:

-Page 13, lines 263-265: “PSMA-PET is considered the imaging of choice to identify PCa lesions in patients with recurrent PCa after primary treatments [18].  also Thus, considering patients who show PSA recurrence after salvage therapies, PSMA-PET may identify thus identifying oligo-recurrent men eligible for MDT”